# Research on Frequent Itemset Mining of Imaging Genetics GWAS in Alzheimer’s Disease

**DOI:** 10.3390/genes13020176

**Published:** 2022-01-19

**Authors:** Hong Liang, Luolong Cao, Yue Gao, Haoran Luo, Xianglian Meng, Ying Wang, Jin Li, Wenjie Liu

**Affiliations:** 1College of Intelligent Systems Science and Engineering, Harbin Engineering University, Harbin 150001, China; lh@hrbeu.edu.cn (H.L.); caoluolong@hrbeu.edu.cn (L.C.); lpl0016@163.com (Y.G.); 13945056105@163.com (H.L.); 2School of Computer Information and Engineering, Changzhou Institute of Technology, Changzhou 213032, China; mengxl@czust.edu.cn; 3School of Computer Science and Engineering, Changshu Institute of Technology, Changshu 215500, China; wangying0129@126.com

**Keywords:** vGWAS, FPM, Eclat, association rules, FI, Alzheimer’s disease

## Abstract

As an efficient method, genome-wide association study (GWAS) is used to identify the association between genetic variation and pathological phenotypes, and many significant genetic variations founded by GWAS are closely associated with human diseases. However, it is not enough to mine only a single marker effect variation on complex biological phenotypes. Mining highly correlated single nucleotide polymorphisms (SNP) is more meaningful for the study of Alzheimer's disease (AD). In this paper, we used two frequent pattern mining (FPM) framework, the FP-Growth and Eclat algorithms, to analyze the GWAS results of functional magnetic resonance imaging (fMRI) phenotypes. Moreover, we applied the definition of confidence to FP-Growth and Eclat to enhance the FPM framework. By calculating the conditional probability of identified SNPs, we obtained the corresponding association rules to provide support confidence between these important SNPs. The resulting SNPs showed close correlation with hippocampus, memory, and AD. The experimental results also demonstrate that our framework is effective in identifying SNPs and provide candidate SNPs for further research.

## 1. Introduction

The brain imaging genetics, as an emerging research field, provides a new approach to study the effect of genetic variations on the brain. The imaging phenotype was regarded as an intermediate phenotype between genetic variants and diagnosis. The imaging genomics combining imaging data and genetic data was applied to explore the pathogenesis of complex diseases, diagnose early diseases, and obtain the phenotypic characteristics of lesions in a multi-modal, high-throughput, and non-invasive manner [1]. Moreover, the relationship between genes and related brain changes can be captured in many studies [2]. Compared with pure genetic research, the combination of brain imaging phenotypes and genetic data is more effective to analyze the genetic variation or assess genetic risks on the brain.

Genome-wide association study (GWAS), proposed by Christopher et al., is a method to find the associations between genetic variations and pathological phenotypes [3]. It combines genetic variations at the single nucleotide polymorphism (SNP) level with imaging phenotype and analyzed the associations between a region of interest (ROI) and SNPs without any prior knowledge of pathology. At present, a large number of GWAS studies have cataloged over 1200 risk alleles for common complex diseases and treats. Stein et al. [4] proposed a voxel based GWAS (vGWAS) method to identify mutations in the entire human genome, reducing the probability of missing important genes and diseased brain regions. The vGWAS was the first voxel based GWAS to find genetic variations associated with brain structure in higher level of refinement. However, these methods were merely useful to find single SNP associated with biological phenotypes [5]. In addition, most of these variants are located in nongenetic regions, and further research is needed to determine whether these variants directly cause the disease through affecting the regulatory factors, or whether they are in a state of linkage disequilibrium with the pathogenic variants.

Since complex diseases were mostly caused by non-linear multiple genetic variations, many methods for multiple SNPs analysis were derived [5,6,7,8]. By combining deep learning stacked autoencoders and association rule mining, the SAERMA (stacked autoencoder rule mining algorithm) method extended GWAS to explore significant SNPs associated with extreme obesity [9]. Sofianita et al. [10] proposed a frequent pattern mining (FPM) algorithm called iterative soft-thresholding (ISTA) algorithm to search frequent itemsets (FIs) of SNPs on the level of individual genotyping data. Even though the studies focusing on detecting meaningful SNP-sets attracted many researchers, there are still limitations on the interpretation of the results [11]. Alzheimer’s disease (AD) is a kind of disease caused by brain lesions and attracts the attention of more and more researchers [12]. To date, a lot of studies imply that structural and functional abnormalities of the brain (e.g., phenotypic, or molecular abnormalities associated with AD) are heritable [13,14,15]. FPM was a problem worthy of intensive study because it was widely used on a series of data mining tasks such as classification, clustering, and outlier analysis [16]. Identifying hidden patterns existing in a dataset was the basic step of constructing association rules for data analysis. All these studies were faced with the memory problems of runtime and computation effectively [17].

In this study, we proposed a framework to identify the SNPs that were highly related to each other and analyzed the correlation of these SNPs with AD related phenotypes. Firstly, to obtain the significance of voxel and SNP, we applied vGWAS to the genotyping data and imaging data of 1515 participants. Then, we applied two algorithms, FP-Growth and Eclat, in this research and used the association rules of hidden patterns sequentially to mine closely connected frequent SNPs. Finally, we analyzed the correlation between identified SNP frequent itemsets (FIs) and hippocampus, memory, and AD. Figure 1 shows the workflow of this research.

## 2. Materials and Methods

### 2.1. Data Source

We downloaded imaging and genotyping data from ADNI (Alzheimer’s Disease Neuroimaging Initiative, adni.loni.usc.edu, accessed on 4 December 2021) dataset firstly. A total of 1515 non-Hispanic white participants had high-quality genotype data and MRI image data in ANDI database at the same time, so they were included in the study after quality control [18] (Table 1). 

Then, MRI images of all 1515 samples were preprocessed with T1-weighted data and standardized according to the Montreal Neurological Institute (MNI) space. Next, we extracted all voxels’ volume using voxel-based morphometry. In short, the scan was aligned with the T1-weighted template image, and divided into gray matter, white matter, and cerebrospinal fluid, and scanned into the MNI space. Then, gray matter density (GMD) was extracted and smoothed using an 8 mm FWHM kernel (182 × 218 × 182 scale). To reduce the calculation time, we down sampled the GMD image into 61 × 73 × 61 (271,633 voxels totally). Finally, 49,900 voxels in the 116 AAL (Automated Anatomical Labeling) Atlas ROIs were chosen for further analysis. The imaging data preprocessing workflow is shown in Figure 2.

### 2.2. Data Processing and Correlation Matrix

In this study, we focused on 20 genes that were significantly associated with AD (Appendix A) from large meta-analysis [19,20,21,22]. SNPs located in ±20 Kbps of these 20 genes were extracted as candidate genetic variants. With the resulting 1784 SNPs (Hardy–Weinberg equilibrium test *p*_HW_ ≥ 10^−6^), vGWAS was performed using linear regression in plink (www.cog-genomics.org/plink/1.9/, accessed on 1 November 2021 [23]). Gender, age, education, and the top 4 principal component analysis results were used as covariates. Then, the correlation matrix between SNPs and voxels was obtained.

Using GWAS results, we obtained the transactional dataset (TD, TD_num = 49,900 voxels) consisted of significant items (here we just set *p*-value ≤ 0.05 as a threshold and ignored the false discovery, for the latter frequency of occurrence could exclude some outliers of SNPs). The support rate of an item $l_{i},\left(i=1,2,\cdots ,1784\right)$ is defined as:(1)$$Support\left(l_{i}\right)=\sum_{j=1}^{TD\_num}Identity\left(P\left(l_{i}\right)\leq 0.05\right)$$
(2)$$Support\_rate\left(l_{i}\right)=\frac{Support\left(l_{i}\right)}{TD\_num}$$

In Equation (1), $Identity\left(.\right)$ is an indicator function. $Support\left(l_{i}\right)$ is defined as the count of transactions that contains the significant item $l_{i}$. In addition, the $Support\_rate\left(l_{i}\right)$ in Equation (2) is the proportion of Equation (1) mentioned transactions to the total TD. Table 2 shows the first 10 frequent SNPs sorted by support rate of voxels on TD.

### 2.3. Frequent Itemsets Mining

One of the urgent problems to solve in scientific research is how to mine meaningful information from massive data rapidly and accurately [16]. Current frequent itemset mining algorithms are summarized into 3 categories: join-based algorithms, tree-based algorithms, and recursive suffix-growth algorithms [24]. In this study, we applied 2 methods, FP-Growth and the Eclat algorithm, separately to identify SNPs and compare their performance and the obtained results.

The FP-Growth [25] was an FPM algorithm with high counting efficiency and the cost of its candidate generation process was relatively low. In the process, a chain of pointers threads was used to store the items in the FP-Tree and these pointers were maintained to form the conditional FP-Tree for an item. The FIs were extracted through the compressed representation. The detailed process is summarized in Table 3.

Equivalence class transformation (Eclat) [26] was an algorithm to mine FIs using the recursive intersection of vertical-form transaction list. Firstly, we obtained the frequent 1-itemsets according to the preset minimum support rate *s*. Subsequently, the frequent (k + 1)-itemsets were generated by integrating the transactions of the frequent k-itemsets. Finally, while all the FIs are different from each other and no other FIs can be found, this repeating process ended. The database was scanned only once even when we want to identify the (k + 1)-itemsets, so the running time was greatly reduced. The detailed process is summarized in Table 4.

### 2.4. Construct Confidence of Frequent Itemsets

We introduced the concepts of association rules and confidence into the FP-Growth algorithm and Eclat algorithm, so that the frequent itemsets provided more information about items (SNPs) in transactions (brain voxels). In order to measure the association between items in an FI, we defined a quantitative index of (*A*→*B*) named confidence as Equation (3).
(3)$$Confidence\left(A\to B\right)=\frac{Support\left(A\cap B\right)}{Support\left(A\right)}$$
where $Confidence\left(A\to B\right)$ represents the confidence from *A* to *B*. $Support\left(A\cap B\right)$ represents the support of *A* and *B*; $Support\left(A\right)$ represents the support of *A*. The relationship between the association rules is determined by confidence, through which we identified closely connected FIs.

We controlled the size of FIs by setting the experimental support rate threshold *s*. Subsequently, we annotated 1-item FIs mined by the Eclat algorithm using SNPnexus (SNP Annotation Tool (snp-nexus.org accessed on 1 November 2021)) [27] based on GRCh38 ensemble resources.

### 2.5. Statistical Analysis of SNPs

To assess the biological significance of identified FIs, we calculated the correlation between FIs and 4 different features closely related to AD (emotional responses, hippocampus, memories, learning task) using NeuroSynth package (https://www.snp-nexus.org/v4/ accessed on 1 November 2021) [28]. This package takes thousands of published articles reporting the results of fMRI studies, Interactive, it contains meta-analyses of 1334 terms, and functional connectivity and coactivation maps for over 150,000 brain locations. We can easily determine the association of a specific MRI image and term by this project.

For the FIs we discovered, epistasis analysis was applied to discover mutual effects between two SNPs (PLINK v1.90b6.18, www.cog-genomics.org/plink/1.9/ accessed on 1 November 2021) [23]. Moreover, if two SNPs were in the same chromosome, we explored the linkage disequilibrium (LD) between them.

## 3. Results

Figure 3 presents the number of FIs for 49,900 voxels in different support rate threshold values. It showed that as the support rate thresholds increased, the number of FIs decreased with the increase of support rate threshold *s*, and the 1-item numbers mined from different algorithms are the same when the support rate threshold is 0.25. This indicates that our mining results are consistent with algorithms.

However, the number of FIs is approximately zero when the threshold is above 0.5. Because the 49,900 voxels were too large, we analyzed two smaller TDs: the right hippocampus (302 voxels) and left hippocampus (281 voxels).

Since we used the smaller TDs, the number of FIs enlarged obviously (Figure 4). The same as Figure 3, Figure 4 also showed that too strict a support rate threshold excluded FIs that contained significant SNPs, while too loose a threshold included a large amount of candidate FIs for testing. Moreover, the FIs number in the left hippocampus was larger than that in the right hippocampus. This demonstrated that the significant SNPs aggregation may exist in the left hippocampus. It is worth noting that the *s* is a support rate of an FI in the full list, out final goal is to find closely connected items in the FIs, so the selection of *s* should enable the later confidence between these frequent items to be relatively high (>0.8 usually).

Table 5 showed the top five k-item FIs sorted by support rate using the Eclat algorithm (the complete results are in Figure 5 and Appendix A). The rs10277969, rs10498633, and rs11731587 were all included in the top five k-item (k = 1, 2, 3, 4, 5) FIs of right hippocampus and left hippocampus. In addition, 10 5-item FIs were found in the left hippocampus totally while zero was found in the right hippocampus.

Since we applied association rules and confidence into the Eclat algorithm, the association rules of top five 2-item FIs in the right hippocampus are shown in Table 6. Notably, the confidence (defined by Equation (3)) of “rs1918296 to rs10277969” is 0.99, which means that for voxels in ROI 37, 99% FIs that contain rs1918296 also contain rs10277969. There is high confidence (≥0.90) in both “rs11731587 to rs1047389” and “rs1047389 to rs11731587”, which indicates a strong relationship between the two items in one FI. A complete description of association rules on FIs is in Appendix A.

We annotated 21 frequent SNPs of right hippocampus and 20 of left hippocampus derived from former mentioned FPM algorithms (Figure 5, Appendix A) using SNPnexus, and the annotation results are shown in Figure 6. The predicted function (Appendix A) of the SNP substitution [29] is based on its first nucleotide location on the transcript. We find about 3/4 of these predicted functional ensemble consequences located in intronic regions, and others in intronic (splice site), 5 utr, 3 utr, 5 upstream, and 3 downstream regions. This indicates that these SNPs play an important role in functions such as transcription and translation.

To assess the biological significance of identified FIs, the correlations between four AD-relating features and FIs were calculated, and the results are shown in Figure 7. We observed that the correlation between “emotional responses” and FIs increases significantly at 3-item in the right hippocampus and then stabilized. It also increases significantly at 2-item in the left hippocampus and then returns to the same as 1-item. For the term “hippocampus” and “memories”, their correlation with the right hippocampus dropped (left hippocampus raised) notably when the item size of FI changed from 1 to 2 and then gradually recovered with the item size increased to 3, 4, and 5. It is worth noting that the feature “learning task” meets a great change of correlation with different item sizes and different ROIs. Comparing with 1-item FI, the correlation between k-item FIs (k = 2, 3, 4) and “learning task” generally decreased, but that of 5-item FIs remained the same as 4-item FIs.

## 4. Discussion

In this study, we performed GWAS by jointly analyzing the genetic and imaging data to explore their associations with AD. Then, two FPM methods (FP-Growth, and Eclat) and association rules of FIs were used to mine multi-SNPs effects which were associated with specific phenotypes. These two FPM methods have mined the same number of FIs, but the Eclat algorithm has the highest mining efficiency and uses the least time (Appendix A), which suits Chee et. al.’s research well [16]. In order to further explore the associations between items in FI, we calculate the confidence and verify the rationality of these FIs. In addition, functional annotations and feature-correlation are used to measure SNPs’ additive influence on brain hippocampus.

From Jansen’s meta-GWAS study [30], seven SNPs were the significant genetic variants (*p*-value ≤ 0.05) among the 21 frequent SNPs of right hippocampus derived from former mentioned FPM algorithms (Figure 5, Appendix A), and the other 14 frequent SNPs had poor correlation with AD. Similarly, we identified12 frequent SNPs that were insignificant in GWAS study (*p*-value ≥ 0.05). Although these SNPs are not closely associated with the overall pathology of AD, they have a wide range of influence on brain structural variation, which are potential therapeutic targets for AD.

The hippocampus, located in the temporal lobe of cerebral cortex, was a cortical region that regulated emotion, learn, motivation, and memory [31]. A lot of research works have demonstrated the pathological effects of the hippocampal structural or functional variation on human aging, AD, and dementia [32,33]. The volume of the hippocampus changes when an individual has severe AD or dementia. So, we analyzed the frequent itemsets in hippocampus in this study and some notable FIs were identified in right and left hippocampus (Appendix A). Here, we discuss some FIs with high frequency.

### 4.1. 1-Item FI: (rs10498633)

The rs10498633 (chr14: 92926952, G>T) is located in the overlapped region of the SLC24A4 and RIN3 gene. The support rate of this FI in the right hippocampus is 0.82 and 0.89 in the left hippocampus, which shows that rs10498633 has a wide range of effects on the two brain regions. Yan et al. [34] found that rs10498633 in SLC24A4 significantly related to the density and length of brain fibers connecting Cerebellum and Somato-Motor, Ventral Attention and Cerebellum, Ventral Attention and Subcortical. In addition, rs10498633 has an important effect on fiber anisotropy, length of fibers and the number of fibers. These three indicators are three methods of brain connection measurement in Alzheimer's disease. Moreover, Jun et al. [35] studied the association between rs10498633 and the gene encoding tau protein. In the research of Tan et al. [36], SLC24A4 and RIN3 were associated with both brain amyloidosis and tauopathy, implying that this SNP directly or indirectly contributes to the risk of AD.

Georgios D. Mitsis et al. [37] proposed a definition that the top 20% voxels activated by SNPs in a single-subject anatomical ROIs was a reliable and sensitive approach to represent region of interest (ROI). For a frequent SNP, the top 20% voxels descending ranked by heritability were kept, and if all their *p*-value were under 0.05, we determined that this ROI was activated by the frequent SNP. Specifically, we counted the effects of a 1-item FI (the rs10498633) on all 116 ROIs and 12 Hippocampus subregions. Table 7 presents the 10 Brain ROIs and 6 Hippocampus subregions activated by rs10498633.

Previous research suggested that olfactory dysfunction in AD was associated with pathological changes of tau protein in the olfactory bulb and olfactory projection area [38,39]. It was also confirmed that AD-related olfactory dysfunction was caused by pathological changes of tau protein [40,41]. The insular cortex was a central brain region characterized by multiple functions and extensive connections [42]. A recent study showed that the insula was crucial in the human brain networks and affected many vulnerable regions of AD [43]. In the review of Huri et.al, they summarized the insular cortex, AD, pathology, and their effects on blood pressure variability [44].

Plenty of research on the activated six hippocampus subregions mentioned in Table 7, Lora et al. [45] founded that Hippocampal-amygdaloid Transition area and Cornu ammonis 1 volume were biomarkers for dissociative amnesia. In the research of Christopher et al. [46], disorder and depression symptom severities were negatively associated with each of HATA, CA2/3, molecular layer, and CA4.

The reduction of hippocampal volume resulted in the memory loss of human, which was a core feature of AD [47,48]. Anna et al. [49] presented the correlations between hippocampal distance and AD using 7T MRI images. The amygdala, which collected pathological proteins, was identified to play a crucial role in human brain as a central communication system. In addition, this was considered to affect the progression and diagnosis of many degenerative diseases, such as AD, chronic traumatic encephalopathy, and Lewy body diseases [50,51]. In the research of Dingailu et al. [52], the fusiform gyrus showed the epigenetic characteristics of AD. Mario et al. [53] reviewed the experimental and humans studies, and summarized the evidence linking temporal epileptiform activity, network hyperexcitability, and AD. In summary, the rs10498633 has an effect on AD pathology in many brain ROIs.

### 4.2. k-Item FI: (k = 2, 3, 4, 5)

The 2-item FI (rs10498633, rs10277969) was found in both the right hippocampus and left hippocampus. Their confidence level exceeds 0.8 (Table 8), indicating a high dependence between the two SNPs. The rs10277969 (chr7: 148035192, A>G) is an Intron Variant of the CNTNAP2 gene. Scharf, J. M., et al. conducted a meta-GWAS analysis of Tourette’s syndrome (TS) and found that the rs10277969 in CNTNAP2 was an important candidate SNP (P = 7.8 × 10^−4^) [54], and CNTNAP2 variants were associated with complex disorders such as depression, schizophrenia, and dyslexia [55]. As shown in Figure 7, we found that when the item size increased from 1 to 2, the correlation growth rates between "learning tasks" and FIs in both regions increased. Nevertheless, the correlation growth rate of “emotional responses” shows opposite trends in the two ROIs. This may be caused by the different operating mechanisms and functions of the two regions.

For the 3-item FI (rs10498633, rs10277969, rs1047389), a new SNP is included (rs1047389, chr4: 11401087, A>G, locate in Synonymous Variant gene HS3ST1). There is no research to prove the influence of this SNP on AD or hippocampal morphological changes. However, when the new SNP "rs1047389" was added to FI, the correlations’ growth rate between the identified FIs and four features (Figure 7) changed greatly, proving that it can greatly affect the AD-related function. Moreover, Nicole et al. [56] found that the HS3ST2 gene, a homologous gene of HS3ST1 [57], plays an important role in the pathology of tau associated with Alzheimer's disease.

The 4-item FI (rs10498633, rs10277969, rs1047389, rs11731587) was expanded from the former mentioned 3-item FI (rs10498633, rs10277969, rs1047389). Its support rate is 0.56 in the right hippocampus and 0.60 in the left hippocampus (Table 5 and Appendix A). Even though there is no direct evidence to support the role of rs11731587 (chr4: 11390069, G>A), the correlations between the identified FIs and four features (Figure 7) prove that the new SNP “rs11731587” can greatly affect the “learning task”, and is negatively correlated with the hippocampus. Therefore, we infer that this 4-item FI has an additive effect and affects the structural features of the hippocampus.

In the 5-item FI (rs10498633, rs10277969, rs1047389, rs11731587, rs2242065), a new phenomenon appears: as the item size rises to 5. Compared with other SNPs, the newly added “rs2242065” (chr15: 58839298, C>T) has no obvious contribution to the growth rate of correlation (Figure 7). In addition, rs2242065 has a low confidence with the previous 4-item FI “rs10498633, rs10277969, rs1047389, rs11731587” (Table 8). We can infer that a larger FI is not always better, and too many SNPs in an FI will weaken the cumulative effect of correlation.

The rs1047389 and rs11731587 were both located in chromosome 4 but not in a same gene region as former mentioned, and the linkage disequilibrium (LD) between them was calculated to be 0.447 (the other three variants are distributed on different chromosomes), excluding the effect of LD and pleiotropism. From the epistasis analysis results in Appendix A, we found a pair of significant epistasis SNPs, rs10498633 × rs10277969, which further demonstrates and verifies the joint effect of the FI we mined.

## 5. Conclusions

In this study, we applied the Frequent itemset mining method into vGWAS and mined a list of frequent SNP sets (rs10498633, rs10277969, rs1047389, rs11731587, rs2242065), which were closely connected and several hippocampus features (i.e., emotional responses, hippocampus, memories, and learning task) concerning Alzheimer’s disease. These closely associated SNPs gave a novel comprehension of the progression and pathology for AD. Moreover, our method provides a novel approach to discover genetic variants that have widespread influence on a range of AD pathologic features.

Due to the interaction between genetic factors and environmental factors, complex diseases have their complexity. As the item size increases, the identified FIs show an additive trend of correlation with AD-related features. However, this trend disappears when an FI contains too many items. There are also some limitations of our work. First, we down sampled the MRI image before conducting GWAS analysis to save computational costs, which may ignore some seemingly unimportant information. Second, although the FIs results are derived from three different mining algorithms; further research is needed to validate their effect in the pathological process of AD.

## Figures and Tables

**Figure 1 genes-13-00176-f001:**
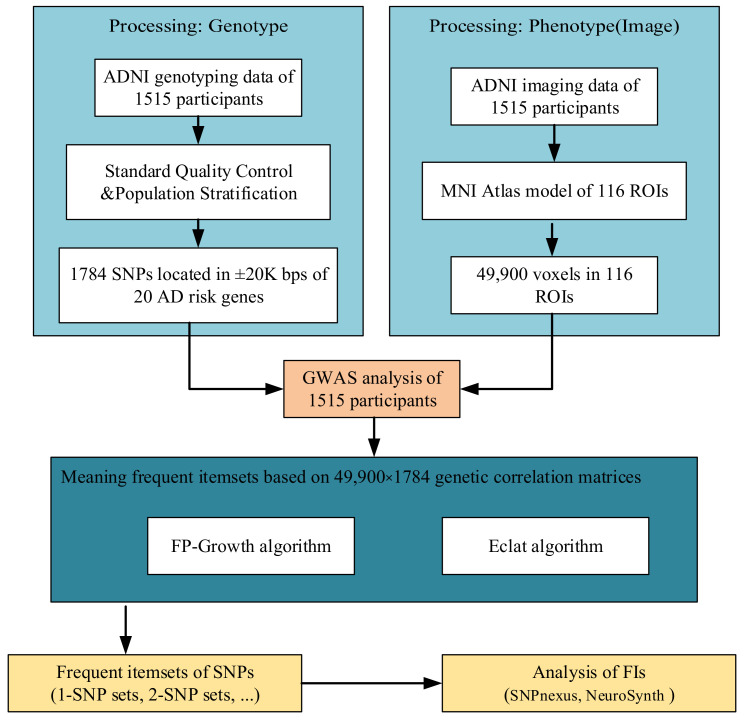
The workflow of this research. ADNI: Alzheimer’s Disease Neuroimaging Initiative; MNI: Montreal Neurological Institute; SNP: single nucleotide polymorphism; GWAS: genome-wide association study; FIs: frequent itemsets.

**Figure 2 genes-13-00176-f002:**
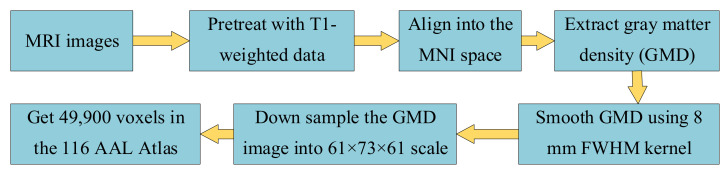
The workflow of imaging data pre-processing.

**Figure 3 genes-13-00176-f003:**
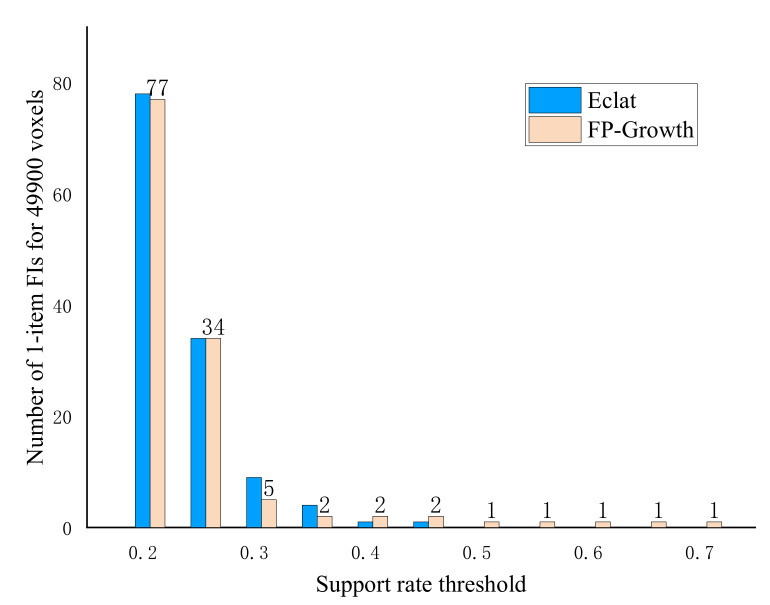
Number of frequent itemsets (FIs) for 49,900 brain voxels in different support rate threshold value using 2 algorithms.

**Figure 4 genes-13-00176-f004:**
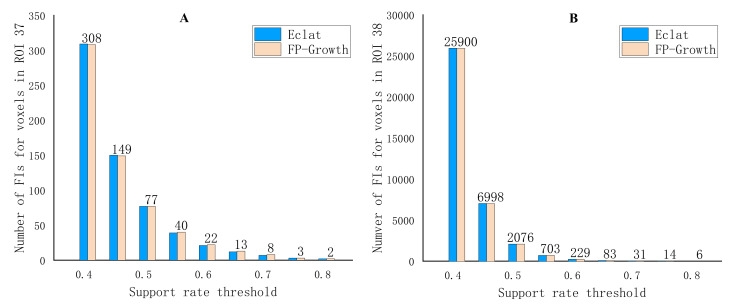
Number of 1-item FIs in different support rate threshold value for right hippocampus (**A**) and left hippocampus (**B**).

**Figure 5 genes-13-00176-f005:**
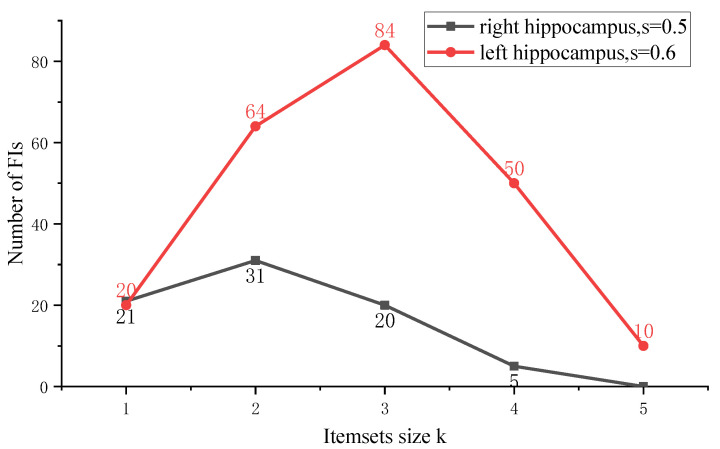
Number of k-item FIs for *s* = 0.5 in right hippocampus and *s* = 0.6 in left hippocampus. FI: frequent itemset.

**Figure 6 genes-13-00176-f006:**
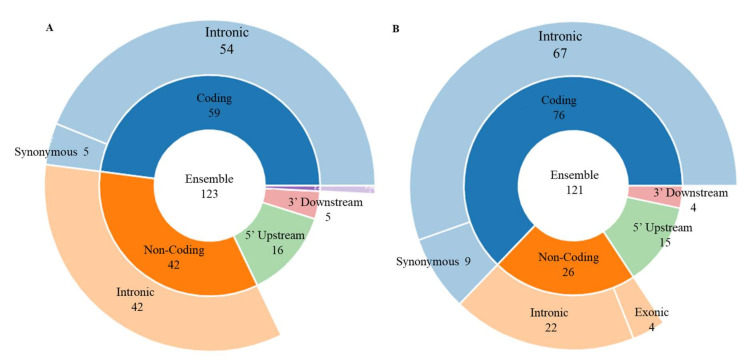
Predicted function of 21 frequent SNPs on right hippocampus (**A**) and 20 frequent SNPs on left hippocampus (**B**).

**Figure 7 genes-13-00176-f007:**
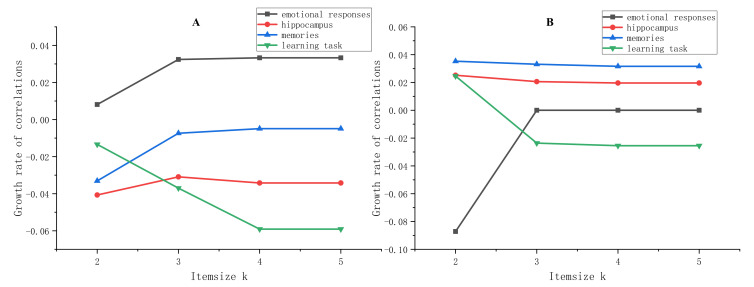
The growth rate of correlations between 4 features and identified FIs in right hippocampus (**A**) and left hippocampus (**B**). The baseline 1-item FI is (rs10498633), 2-item FI is (rs10498633, rs10277969), 3-item FI is (rs10498633, rs10277969, rs1047389), 4-item FI is (rs10498633, rs10277969, rs1047389, rs11731587), 5-item FI is (rs10498633, rs10277969, rs1047389, rs11731587, rs2242065). Using 1-item as the benchmark, the growth rate of k-item (k = 2, 3, 4, 5) relative to 1-item was calculated.

**Table 1 genes-13-00176-t001:** Demographic statistics of the participants in ADNI database.

Characters	CN	SMC	EMCI	LMCI	AD
Number of samples	353	89	273	504	296
Gender(M/F)	187/166	36/53	153/120	309/195	166/130
Age (year, Mean ± SD)	74.9 ± 5.7	72.2 ± 5.7	71.3 ± 7.1	74.0 ± 7.6	74.7 ± 7.6
Education (year, Mean ± SD)	16.1 ± 2.7	16.8 ± 2.6	16.1 ± 2.6	16.0 ± 2.9	15.5 ± 2.9

CN: clinically normal; SMC: subjective memory concerns; EMCI: early mild cognitive impairment; LMCI: late mild cognitive impairment; AD: mild Alzheimer’s disease dementia.

**Table 2 genes-13-00176-t002:** Top 10 frequent SNPs sorted by coverage rate on 49,900 brain voxels.

NO.	SNP	Support Rate
1	rs6014724	0.46
2	rs11731587	0.39
3	rs7806	0.36
4	rs6024860	0.35
5	rs1060743	0.34
6	rs4243693	0.34
7	rs7790238	0.33
8	rs7219391	0.31
9	rs386274	0.30
10	rs6092321	0.30

**Table 3 genes-13-00176-t003:** The FP-Growth algorithm process. (FP-Tree: FPT, current itemset suffix: P = *ϕ*, Support rate threshold: *s*).

Begin:
If (FPT is a single path or empty):
For each subset of item in path (return FI and its support judge by s)
Else:
(
For each item i in chain of pointers
(
Generate conditional pattern base Pi = (*i*) ∪ P and get its support
Extract conditional FP-tree FPT_i_ from chain of pointers in P_i_
If (FPT*_i_* ≠ ∅) recursion FP-Growth (FPT*_i_*, P*_i_*, s)
)
)
end

**Table 4 genes-13-00176-t004:** The Eclat algorithm process. (frequent pattern itemset: FP, Support rate threshold: *s*).

Begin:
$\mathrm{For}\;\mathrm{each}\;\mathrm{item}\;l_{i}$ in FP (
FP*_i_* = ∅
$\mathrm{For}\;\mathrm{each}\;\mathrm{item}\;l_{i}$ in FP and *j* > *i* (
$l_{ij}=l_{i}\cap l_{j}$
$Itemset\left(l_{ij}\right)=Itemset\left(l_{i}\right)\cap Itemset\left(l_{j}\right)$
$\mathrm{If}\;(Support\_rate\left(l_{ij}\right)\geq s$$):\;\mathrm{add}\;l_{ij}$ into FP_i_
Recurve Eclat(FP*_i_*, *s*)
)
)
end

**Table 5 genes-13-00176-t005:** k-item FIs ordered by support rate using Eclat algorithm. FI: frequent itemset.

Right Hippocampus		Left Hippocampus	
3-Item, 4-Item and 5-Item FIs (Top 5)	Support Rate	3-Item, 4-Item and 5-Item FIs (Top 5)	Support Rate
rs1047389, rs11731587, rs10277969	0.65	rs10277969, rs2242065, rs10498633	0.72
rs1047389, rs10498633, rs10277969	0.63	rs10277969, rs2242065, rs1047389	0.71
rs1047389, rs11731587, rs16881446	0.60	rs2242065, rs10498633, rs1047389	0.71
rs11731587, rs10498633, rs10277969	0.58	rs7563345, rs10498633, rs1047389	0.70
rs1047389, rs11731587, rs10498633	0.58	rs2242065, rs10498633, rs6082	0.70
rs1047389, rs11731587, rs10498633, rs10277969	0.56	rs10277969, rs2242065, rs10498633, rs6082	0.67
rs1047389, rs11731587, rs16881446, rs10277969	0.54	rs10277969, rs2242065, rs10498633, rs1047389	0.67
rs1047389, rs10277969, rs1918296, rs886969	0.53	rs7563345, rs2242065, rs10498633, rs6082	0.65
rs1047389, rs11731587, rs10277969, rs1918296	0.52	rs10277969, rs2242065, rs10498633, rs7563345	0.65
rs1047389, rs10498633, rs10277969, rs1918296	0.51	rs10277969, rs2242065, rs7000615, rs1047389	0.65
NULL	NULL	rs10277969, rs7563345, rs2242065, rs10498633, rs6082	0.63
		rs10277969, rs1047389, rs2242065, rs10498633, rs6082	0.62
		rs10277969, rs1047389, rs7563345, rs2242065, rs10498633	0.62
		rs10277969, rs1047389, rs2242065, rs10498633, rs7000615	0.61
		rs1047389, rs7563345, rs2242065, rs10498633, rs6082	0.61

**Table 6 genes-13-00176-t006:** Association roles of 2-item FIs (top 5) and corresponding confidence.

2-Item FIs (Top 5) in ROI 37	Confidence	2-Item FIs (Top 5) in ROI 37	Confidence
rs10498633 to rs10277969	0.90 (0.74/0.82)	rs10277969 to rs10498633	0.84 (0.74/0.88)
rs1047389 to rs10277969	0.94 (0.74/0.79)	rs10277969 to rs1047389	0.84 (0.74/0.88)
rs11731587 to rs1047389	0.97 (0.71/0.73)	rs1047389 to rs11731587	0.90 (0.71/0.79)
rs11731587 to rs10277969	0.92 (0.67/0.73)	rs10277969 to rs11731587	0.76 (0.67/0.88)
rs10277969 to rs1918296	0.76 (0.67/0.88)	rs1918296 to rs10277969	0.99 (0.67/0.68)

**Table 7 genes-13-00176-t007:** Brain ROIs and Hippocampus subregions activated by rs10498633.

Activated Brain ROIs:		Activated Hippocampus Subregions:	
NO.	ROI	NO.	Subregion
1	Frontal_Inf_Orb	1	Hippocampal-amygdaloid Transition area
2	Olfactory	2	Cornu ammonis 1
3	Insula	3	Pre subiculum
4	Hippocampus	4	Cornu ammonis 4
5	Para Hippocampal	5	Para subiculum
6	Amygdala	6	Hippocampal fissure
7	Fusiform		
8	Temporal_Pole_Sup		
9	Temporal_Pole_Mid		
10	Temporal_Inf		

**Table 8 genes-13-00176-t008:** Association rules of some significant FIs.

Association Rules	Confidence	
	Right Hippocampus	Left Hippocampus
rs10498633 to rs10277969	0.90	0.86
rs10498633, rs10277969 to rs1047389	0.85	0.91
rs10498633, rs10277969, rs1047389 to rs11731587	0.88	0.87
rs10498633, rs10277969, rs1047389, rs11731587 to rs2242065	-- *	--

* The corresponding confidence is under 0.7.

## Data Availability

The data is available at http://adni.loni.usc.edu/, accessed on 7 October 2020.

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
