# Peer review of "Research on Frequent Itemset Mining of Imaging Genetics GWAS in Alzheimer’s Disease"

_genes, 2022, doi:10.3390/genes13020176_

Round 1

Reviewer 1 Report

In the present study, the authors used two frequent pattern mining framework to analyze the GWAS results of functional magnetic resonance imaging phenotypes. They calculated the conditional probability of identified SNPs and have obtained the corresponding association rules to provide support confidence between these important SNPs. They performed correlations of these SNPs with hippocampus, memory, and Alzheimer's disease occurrence. However, there are still some issues in the manuscript and that are needs to be addressed properly .

  1. Authors need to provide separate paragraph for the statistical analysis they did in the present study under a separate subheading.
  2. How the authors confirmed that the SNPs present in the gene, they did select for the present analysis are in Hardy-Weinberg Equilibrium.
  3. In the last paragraph of methodology, section author mentioned that they explored LD analysis. But as far as I have seen the LD analysis data that have been executed by them in Supplemental table 6 is not correct. They must provide LD data through LD map in LD units with discriminating blocks of conserved LD that have additive distances and locations. Further, they need to provide the allele frequency effect as well.
  4.  Author mentioned that they explored LD analysis. Therefore I recommend providing haplotype analysis data for the SNPs present in the same gene to explore the combined effects of several single nucleotide polymorphisms with linkage disequilibrium.

Author Response

Response to Reviewer 1 Comments

Thank you for your comments concerning our manuscript entitled “Research on Frequent Itemset Mining of Imaging Genetics GWAS in Alzheimer's Disease” (ID: genes-1517788). The reviews are very helpful for us to revise and improve the paper. We have studied the comments carefully and have made corrections and changes accordingly. We hope this revised version is satisfactory (revised parts are marked in red). The follow is an outline of our responses and revisions.

Point 1: Authors need to provide separate paragraph for the statistical analysis they did in the present study under a separate subheading.

Response 1: Thanks for your good advice. We added a separate subheading (Statistical analysis of SNPs) as the section 2.5 for the statistical analysis. Line 163.

Line 163: 2.5. Statistical analysis of SNPs

Point 2: How the authors confirmed that the SNPs present in the gene, they did select for the present analysis are in Hardy-Weinberg Equilibrium.

Response 2: This is an important idea. The Hardy-Weinberg Equilibrium was applied in the pre-processing of the genotype data, and the SNPs selected were satisfied with pHW>10E-6. We added the description of Hardy-Weinberg Equilibrium. Line 108.

Line 108: With the resulting 1,784 SNPs (Hardy-Weinberg equilibrium test pHW≥10E-6), vGWAS was performed using linear regression in plink.

Point 3: In the last paragraph of methodology, section author mentioned that they explored LD analysis. But as far as I have seen the LD analysis data that have been executed by them in Supplemental table 6 is not correct. They must provide LD data through LD map in LD units with discriminating blocks of conserved LD that have additive distances and locations. Further, they need to provide the allele frequency effect as well.

Response 3: We agree. We revised our experiment and found that there was no linkage influence among the five SNPS. And there was no influence on our conclusion. However, our results indicate that the potential relationship among these SNPs may affect the hippocampus. Therefore, we deleted the supplementary table 6, and revised the description in line 352-357 and line 171-174.

Line 171-174: For the FIs we discovered, epistasis analysis was applied to discover mutual effects between two SNPs (PLINK v1.90b6.18, www.cog-genomics.org/plink/1.9/, [23]). Moreover, if two SNPs were in the same chromosome, we explore linkage disequilib-rium (LD) between them.

Point 4: Author mentioned that they explored LD analysis. Therefore I recommend providing haplotype analysis data for the SNPs present in the same gene to explore the combined effects of several single nucleotide polymorphisms with linkage disequilibrium.

Response 4: Thanks. We agree with your idea. According to the inspection, only rs1047389 and rs11731587 are located in the same chromosome. The distance between them is 11018 BP (base pair) and not in the same gene fragment. The R2 calculated of LD is 0.447, and there is no potential epistasis relationship between them. Therefore, we revised the description in line 352-357.

Line 352-357: The rs1047389 and rs11731587 were both located in chromosome 4 but not in a same gene region as former mentioned, and the linkage disequilibrium (LD) between them was calculated to be 0.447 (the other three variants are distributed on different chromosomes), excluding the effect of LD and pleiotropism. From epistasis analysis results in Supplementary TABLE 6, we found a pair of significant epistasis SNPs, rs10498633 × rs10277969, this further demonstrate and verify the joint effect of the FI we mined.

We appreciate for Editors’ and Reviewer’s warm work earnestly, and hope that the correction will meet with approval.

Once again, thank you very much for your comments and suggestions.

Reviewer 2 Report

In this article the author the authors examined genome wide association study (GWAS), which is identify the association between genetic variation and pathological phenotype in Alzheimer’s disease patients. Overall, the study is interesting and well written and present. 

  1. First paragraph of the introduction is difficult to understand it should be changed
  2. Why only non-Hispanic white participant include in the study
  3. On which basis 1515 participant include in the study
  4. In each section the abbreviation should be defined
  5. Figure legends should be included
  6. Discussion need improvements

Author Response

Thank you for your comments concerning our manuscript entitled “Research on Frequent Itemset Mining of Imaging Genetics GWAS in Alzheimer's Disease” (ID: genes-1517788). The reviews are very helpful for us to revise and improve the paper. We have studied the comments carefully and have made corrections and changes accordingly. We hope this revised version is satisfactory (revised parts are marked in red). The follow is an outline of our responses and revisions.

Point 1: First paragraph of the introduction is difficult to understand it should be changed.

Response 1: Thanks for your serious review. We agree. We revised paragraph 1 in line 31-40.

Line 31-40:The brain imaging genetics, as an emerging research field, provides a new approach to study the effect of genetic variations on the brain. The imaging phenotype was regarded as an intermediate phenotype between genetic variants and diagnosis. The imaging genomics combining imaging data and genetic data was applied to explore the pathogenesis of complex diseases, diagnose early diseases, and obtain the phenotypic characteristics of lesions in a multi-modal, high-throughput, and non-invasive manner [1]. Moreover, the relationship between genes and related brain changes can be captured in many studies [2]. Compared with pure genetic research, the combination of brain imaging phenotypes and genetic data is more effective to analyze the genetic variation or assess genetic risks on the brain.

Point 2: Why only non-Hispanic white participant include in the study.

Response 2: This is an important idea. To exclude the heterogeneity of genetic loci information caused by ethnic differences,we downloaded the data of non-Hispanic white participants from ADNI.

Point 3: On which basis 1515 participant include in the study

Response 3: We agree. The number of samples with both high-quality genotype data and MRI image data in ANDI database was 1515. We revised the line 84-86 of manuscript.

Line 84-86: A total of 1,515 non-Hispanic white participants had high-quality genotype data and MRI image data in ANDI database at the same time, so they were included in the study after quality control [18] (TABLE 1).

Point 4: In each section the abbreviation should be defined.

Response 4: Thanks for your advice. This have been proofread and revised.

Point 5: Figure legends should be included.

Response 5: Thanks. This have been proofread and revised.

Point 6: Discussion need improvements.

Response 6: We revised our experiment and found that there was no linkage influence among the five SNPS. However, our results indicate that the potential relationship among these SNPs may affect the hippocampus. According to the inspection, only rs1047389 and rs11731587 are located in the same chromosome. The distance of them is11018 BP and not in the same gene fragment. The R2 calculated of LD is 0.447, and there is no potential epistasis relationship between them. Therefore, we deleted the supplementary table 6, and revised the description in line 352-357.

Line 352-357: The rs1047389 and rs11731587 were both located in chromosome 4 but not in a same gene region as former mentioned, and the linkage disequilibrium (LD) between them was calculated to be 0.447 (the other three variants are distributed on different chromosomes), excluding the effect of LD and pleiotropism. From epistasis analysis results in Supplementary TABLE 6, we found a pair of significant epistasis SNPs, rs10498633 × rs10277969, this further demonstrate and verify the joint effect of the FI we mined.

We appreciate for Editors’ and Reviewer’s warm work earnestly, and hope that the correction will meet with approval.

Once again, thank you very much for your comments and suggestions.

Reviewer 3 Report

I would like to thank the Editors and the Authors of the Manuscript "Research on Frequent Itemset Mining of Imaging Genetics GWAS in Alzheimer's Disease" for the opportunity to read and comment this work, which I find very innovative and interesting in its approach towards finding SNPs related to Alzheimer's disease through a combination of genetic data and imaging.

I would like to point out that the caption for Figure 2 in the main text does not correspond to what the figure is representing and must be corrected. 

It is very interesting that you could find 5 SNPs that have no linkage disequilibrium but potential epistatic relationships with each other and influence the activity and processes of the left and right hippocampus.

In the context of your study, how do you interpret the fact that some of the substitutions that you present in Supplementary Tables 4.1 and 4.2 are synonymous, so they do not change the structure and function of the coded protein?

Author Response

Thank you for your comments concerning our manuscript entitled “Research on Frequent Itemset Mining of Imaging Genetics GWAS in Alzheimer's Disease” (ID: genes-1517788). The reviews are very helpful for us to revise and improve the paper. We have studied the comments carefully and have made corrections and changes accordingly. We hope this revised version is satisfactory (revised parts are marked in red). The follow is an outline of our responses and revisions.

Point 1: I would like to point out that the caption for Figure 2 in the main text does not correspond to what the figure is representing and must be corrected.

Response 1: Thanks for your serious review. We revised the caption for Figure 2. Line 113.

Line 113:Figure 2. The workflow of imaging data pre-processing.

Point 2: In the context of your study, how do you interpret the fact that some of the substitutions that you present in Supplementary Tables 4.1 and 4.2 are synonymous, so they do not change the structure and function of the coded protein?

Response 2: This is an important idea. The synonymous coding SNP will not change the protein sequence itself, because not all codon changes will lead to a difference in the amino acid sequence. But this does not mean that these SNPs have no effect on the disease. Relevant studies showed that the effect mainly came from the mRNA secondary structure, protein folding and cell positioning.

Although this nonsense mutation does not change the composition of amino acids, due to the codon preference of protein translation, the process of changing from commonly used codons to less commonly used codons occurs when the ribosome passes through the mRNA fragments around the SNP. However, the folding process in the cell is generally considered to be synchronized with translation. Therefore, these SNPs will affect the time of protein folding and its transfer to the cell membrane, thereby changing the interaction with the substrate and inhibitor.

We appreciate for Editors’ and Reviewer’s warm work earnestly, and hope that the correction will meet with approval.

Once again, thank you very much for your comments and suggestions.
